# Optimization of a Lambda-RED Recombination Method for Rapid Gene Deletion in Human Cytomegalovirus

**DOI:** 10.3390/ijms221910558

**Published:** 2021-09-29

**Authors:** Estéfani García-Ríos, Julia Gata-de-Benito, Mireia López-Siles, Michael J. McConnell, Pilar Pérez-Romero

**Affiliations:** National Center for Microbiology, Instituto de Salud Carlos III Majadahonda, 28220 Madrid, Spain; egarcia@isciii.es (E.G.-R.); juliagatadebenito@gmail.com (J.G.-d.-B.); mireia.lopez@isciii.es (M.L.-S.); michael.mcconnell@isciii.es (M.J.M.)

**Keywords:** cytomegalovirus, BAC, Lambda-RED system, recombination, gene-deletion mutants

## Abstract

Human cytomegalovirus (HCMV) continues to be a major cause of morbidity in transplant patients and newborns. However, the functions of many of the more than 282 genes encoded in the HCMV genome remain unknown. The development of bacterial artificial chromosome (BAC) technology contributes to the genetic manipulation of several organisms including HCMV. The maintenance of the HCMV BAC in E. coli cells permits the rapid generation of recombinant viral genomes that can be used to produce viral progeny in cell cultures for the study of gene function. We optimized the Lambda-Red Recombination system to construct HCMV gene deletion mutants rapidly in the complete set of tested genes. This method constitutes a useful tool that allows for the quick generation of a high number of gene deletion mutants, allowing for the analysis of the whole genome to improve our understanding of HCMV gene function. This may also facilitate the development of novel vaccines and therapeutics.

## 1. Introduction

Human cytomegalovirus (HCMV) is a highly prevalent herpesvirus (HHV-5) that infects approximately 60% of adults in developed countries, while it can reach up to 100% in developing countries [1,2,3]. While CMV infection does not cause complications in immunocompetent individuals [4,5,6], in individuals with an immature immune system and in immunocompromised individuals [7], especially HIV-infected patients and transplant recipients, infection can be associated with severe symptoms [8,9,10,11,12].

HCMV is the largest virus able to infect humans with a genome of approximately 236 Kb [4,13,14,15,16,17]. The genome is a linear, double-stranded DNA molecule consisting of a unique long (UL) and a unique short (US) domain flanked by terminal repeated sequences (TRL and TRS) and by internal repeats (IRL and IRS) resulting in a TRL–UL–IRL–IRS–US–TRS genome organization [18]. It was recently demonstrated that the HCMV genome presents more than 751 translated open reading frames (ORFs) [18,19,20] encoding for 282 viral transcripts, with 206 unique coding potentials [21]. Although many HCMV genes involved in the viral cycle during infection have been characterized [22], the functions of many of the genes encoded in the HCMV genome remain unknown [22].

Bacterial artificial chromosomes (BACs) are in vitro assembled DNA plasmid molecules containing chromosomes that can be stably maintained in *Escherichia coli* [23,24]. BACs can be used for transfecting eukaryotic cells and genome sequencing and for the functional characterization of genomes, especially large viral genomes such as HCMV [25]. The use of BACs overcomes the problems of recombinant methods in cell cultures by applying the powerful techniques developed in bacterial genetics [26]. The HCMV genome integrated in a BAC was previously genetically modified using different approaches such as homologous recombination or transposon-based techniques [23,27,28,29,30,31,32]. During recent years, several studies disrupted HCMV genes by using different methods [24,28,31,32,33,34] such as transposon mutagenesis [35,36], allelic exchange [32,36], CRISPR-based techniques [37,38], small interfering RNAs (siRNAs) [39] and Lambda Red-mediated linear recombination [40,41,42,43,44,45,46,47]. Most of the previous studies generated gene deletion mutants of a small number of genes but did not analyze or optimize the efficiency of the method to be applied for high throughput genome analysis.

The rapid generation of HCMV mutants with high-efficiency and easy-to-use techniques has the potential to improve our knowledge of viral biology, thus facilitating the design of new therapeutic and preventive approaches against HCMV. We used a method for Bacterial Artificial Chromosome Lambda-Red Recombination, previously described by Wanner and Datsenko (2000) [48]. This strategy is based on the replacement of the chromosomal gene sequence with a selectable antibiotic resistance gene that includes the lambda recombination genes *exo*, *bet* and *gam* expressed from the lambda promoter preventing the degradation of the linear DNA and facilitating homologous recombination [49,50]. In the present work, we modified and optimized this method to disrupt single HCMV genes rapidly in a simple and highly efficient manner, which allows for the deletion of a high number of genes from the genome in a small period of time that can be useful for high throughput genome analysis. 

## 2. Results

### 2.1. Recombinant HCMV BAC Construction and Efficiency of the Optimized Method

To test the efficiency and performance of this method, we selected twenty-two HCMV genes for deletion in the HCMV BAC construct (Table 1). Oligonucleotides used to amplify the Kan^R^ gene, including a 20 bp sequence homologous to the Kan^R^ cassette from the pKD4 plasmid and a 50 bp sequence homologous to the 5′ and 3′ UTRs of each of the deleted genes, are shown in Table 2. Either freshly made or frozen *E. coli* DH5α λ pir+HCMV-BAC-pKD46 competent cells were electroporated with the linear PCR fragment containing the Kan^R^ gene flanked by the sequence tails homologous to the gene to be deleted. The target gene Kan^R^ cassette was sequenced, and no mutations were detected. Using the optimized method, we were able to obtain recombination events successfully for all 22 selected genes. To obtain visible transform colonies, the LB-Kan plates needed to be incubated overnight at 37 °C followed by 24–48 h incubation at room temperature. Individual colonies were propagated in kanamycin-supplemented media for further analysis. Results obtained for the deletion of UL6 and UL8 genes are shown as an example. The electroporation parameters were 1.8 KW for both UL6 and UL8, and the time constants (ms) were 5.1 and 5, respectively. 

The efficiency parameters of the optimized method, including the number of tested colonies and the percentage of positive clones for each of the assayed genes, are shown in Table 1. Although the number of colonies obtained for each gene was highly variable, we obtained positive colonies for all the deleted genes, with a percentage of positive colonies that ranged from 1.8 to 100% of the tested colonies. We found no association between the number of positive mutants obtained and the length of the deleted gene (Appendix A). 

We also compared, in parallel experiments, the efficacy of transforming freshly prepared versus frozen electrocompetent cell stocks in seven of the assayed genes (Table 3). Both cell stocks were prepared from the same batch and were electroporated with the same amount of DNA. Surprisingly, six out of the seven tested genes had an increased efficiency in the percentage of positive colonies when using frozen cells. The number of total transform colonies obtained after transformation was also higher when using frozen cells, except for gO and pp65 (Table 3). Notably, the values were extremely variable and gene-dependent. 

### 2.2. Verification of the Deletion of HCMV BAC Constructions

To verify gene disruption and insertion of the linear PCR fragment containing the Kan^R^ gene, all of the constructs were verified using PCR with different oligonucleotide combinations. BAC DNA was isolated from the transformant *E. coli* colonies and analyzed using PCR. PCR amplification was performed using primers homologous to the 5′ and 3′ UTRs of the deleted genes and to an internal region of the Kan^R^ gene (Figure 1A and Appendix A). The parental BAC was used as a control to amplify the corresponding wild type genes. The obtained PCR products were consistent with the predicted size for the deleted genes (Appendix A). Representative results obtained using UL6 (1342 bp) and UL8 (831 bp) genes are shown in Figure 1B. Using ΔUL6-HCMV-BAC and ΔUL8-HCMV-BAC mutant BAC and oligonucleotides upstream and downstream of the target gene, the PCR products obtained were similar for both UL6 and UL8 genes due to the insertion of the Kan^R^ gene, 2117bp and 2092 bp, respectively (Figure 1B lane 2). The PCR products obtained were consistent with the predicted size of the deleted genes.

Two other parallel verification tests were performed to confirm insertion in the HCMV genome. First, a PCR amplification using a 5′ primer upstream of the target gene and a 3′ primer in the Kan^R^ cassette was performed. As shown in Figure 1B, the amplified fragments of around 1.1 Kb were similar for both mutant constructs (Figure 1B, lane 3 and Appendix A). Second, a PCR amplification including a 5′ primer in the Kan^R^ cassette and a 3′ primer downstream of target gene was performed. As shown in Figure 1B, the amplified fragments of around 1.2 Kb were similar for both mutant constructions (Figure 1, lane 4). These results confirmed UL6 and UL8 gene deletion and indicated that the Kan^R^ cassette was inserted in the correct genomic region. Representative images of PCR amplification products obtained to confirm the gene deletion of other ORFs are included in Appendix A.

In addition, in order to test the DNA integrity of the BAC, WT and the BAC from eight of the colonies obtained from one of the rounds of recombination during the deletion of UL6, ORFs were digested with *Eco*RI and *Bam*HI (Appendix A). Both enzymes are commonly used for restriction analysis since they produce evenly spaced fragments upon agarose gel electrophoresis and are useful for detecting any gross alterations of the viral genome. The positive clones for ΔUL6 (lanes 3 and 4 of Appendix A) had a restriction pattern similar to that obtained for the WT BAC. In addition, four of the tested colonies presented an altered restriction pattern compared with the wild type (lanes 3 and 4 of Appendix A). Further validation amplification confirmed that these four colonies were negative since the fragment amplified was similar to WT (Appendix A), suggesting that the Kan^R^ cassette was inserted in another genomic region.

### 2.3. Recombinant Virus Production and PCR Verification

In order to test whether mutant viral progeny could be generated, the ΔUL6-HCMV-BAC and ΔUL8-HCMV-BAC DNA preparations were transfected into MRC-5 cells. As shown in Figure 2 that represents confocal images of the indicated transfections, the GFP signal was visible 7 days after transfection, and the cytopathic effect was observed 2 weeks post-transfection.

The resulting viral progeny was collected from the supernatant of the transfected cells and used for infecting freshly seeded MRC-5 cells in 96-well plates. Fluorescent infected cells were visible 6 days post-infection. To verify the deletion of the indicated gene in the viral progeny, total DNA was isolated fourteen days post-infection from the WT-, ΔUL6-HCMV- and ΔUL8-HCMV-infected cells and used for PCR amplification (Figure 2D). The PCR products obtained were consistent with the expected size for the wild type and both ΔUL6-HCMV and ΔUL8-HCMV mutants. 

## 3. Discussion

In spite of increasing knowledge of the HCMV life cycle obtained over the last decade, the function of an important number of HCMV genes is still unknown [18,22,25]. A possible approach to study the function of a given protein is to generate mutant viruses for the target gene through genetic manipulation of the viral genome. Cloning of the HCMV genome as BAC DNA has become an important tool that allows faster and more efficient genetic manipulation of the genome [23,26,48]. The availability of a HCMV deletion gene collection may contribute to the analysis from different points of view of the functionality and cellular implications of each of the studied genes. 

Linear recombination sharing short homologies has become the method of choice to modify viral BAC genomes genetically due to its efficiency and versatility. Despite the advantages, recombineering also has some limitations [49]. Due to the small size of homologous regions that recombine, some genomic regions containing repetitive sequences can be problematic. In addition, as most homologous recombination-based methods employ PCR-amplified products as template DNA, the PCR products may acquire mutations that could be detected by sequencing the target gene after the recombination. In addition, finally, the sequence of the target gene to be modified in the genome must be known and well annotated. However, a vast number of genome sequences are increasingly available in public databases. 

The Lambda-Red method based on the expression of the Red genes contained in the low copy number plasmid pKD46 [48] facilitates the generation of mutant genomes by deleting individual genes. In addition, the pKD46 plasmid exhibits temperature-sensitive replication and can be easily eliminated by growing the bacteria at 37 °C, and the Red system is under the control of an arabinose-dependent inducible promoter, which prevents undesired recombination events under non-inducing conditions [51]. Gene deletion by homologous recombination involves the exchange between two DNA sequences carried out by recombination systems that require a minimal length of DNA homology [52]. This method is time-consuming and with highly variable efficiency.

The Lambda-Red methodology has been successfully applied to several viral systems such as the baculovirus [53,54], vaccinia [55], Epstein-Barr [56] or MERS-CoV [57]. Generating gene deletions in large viral genomes (such as HCMV) continues to be a challenge that limits the experimental characterization of the gene function in these viruses. In the present study, we optimized the Lambda-Red method to increase its efficiency. Some modifications were performed in the protocol compared to the original study [48] in order to increase efficiency. Firstly, the growth and the arabinose induction were conducted in two separate steps. Thus, the OD600 to make the electrocompetent cells was 1.0 instead of 0.6 used in the original work. In addition, we used 250 ng of the PCR product, while previous works employ 10–100 ng. With this optimization, we were able to obtain at least one positive clone for all the twenty-two tested genes, demonstrating the reproducibility and reliability of the methodology, enabling the generation of a high number of mutant genes of the HCMV genome, decreasing the time employed, which can facilitate the high throughput analysis of multiple protein functions. In addition, this protocol could potentially be used to introduce point mutations in the genome, taking into account some modifications during the cassette design. 

Our results demonstrate that the quality and quantity of the PCR product are crucial as well as the OD600 of the cells for obtaining high recombination efficiency and producing a high number of recombinant cells. We were able to obtain gene deletion mutants for both structural and non-structural proteins in all the assayed genes and the subsequent insertion of the Kan^R^ gene in the desired locus, which also reflect the high efficiency and reproducibility of the methods using the optimized conditions. Additionally, preparation of the competent cells containing the HCMV target genome is also a key factor for recombination success. In most studies [31,48], authors recommend the use of freshly made electrocompetent cells based on a decrease in efficacy when using frozen electrocompetent bacterial stocks [23]. However, using frozen electrocompetent bacterial stocks allows for the previous preparation of large cell stocks that can be stored at −80 °C and ready to be used when necessary. Using frozen stocks increases the reproducibility of the process, decreasing the protocol timing and facilitating the production of the deletion mutants, since the preparation of competent cells is time consuming, which is also an advantage in terms of time and the number of processed samples.

In summary, we optimized the Lambda-Red recombineering method for the generation of gene deletion mutants of the HCMV genome that results in mutant viral progeny. This protocol allows for the deletion of a high number of genes from the genome in a small period of time that can be useful for high throughput genome analysis. The innovations introduced in our study will provide a valuable resource for researchers aiming to generate gene deletions efficiently in large viral genomes.

## 4. Materials and Methods 

A summarized scheme of the methodology for the recombination system is shown in Figure 3. 

### 4.1. Bacteria Strains, Virus, Cells and Media 

The *Escherichia coli* DH5α λ pir strain containing the CMV BADrUL131-Y4 BAC was used for homologous recombination [32]. The BACmid used in this study, containing the CMV strain derived from the AD169 strain in which the UL131 sequence was repaired and containing a chloramphenicol resistance gene, was kindly provided by Dr. Shenk (Princeton University, Princeton, NJ, USA). One crucial aspect for this recombination system is to use a λ pir strain since the recombineering helper plasmid (pKD46) has conditional (oriRγ) replicons that require the trans-acting II protein (the pir gene product) for replication. The pDK4 plasmid containing the kanamycin resistance gene and the pKD46 plasmid containing the ampicillin resistance gene were obtained from Wanner [48]. The pKD46 contains the Exo, β and γ genes expressed under the arabinose inducible promoter involved in homologous recombination [48]. Antibiotic selection of the bacterial strains was performed on LB media or LB agar media (Conda, Torrejon de Ardoz, Madrid, España) supplemented with 100 µg/mL Ampicilin (Amp, A0166-5G, Sigma, Saint Louis, MO, USA), 50 µg/mL Kanamycin (Kan, ThermoFisher Scientific, Waltham, MA, USA) and 34 µg/mL Chloramphenicol (Chl, C0378-5G, Sigma, Saint Louis, MO, USA). The SOC medium (Invitrogen-ThermoFisher Scientific, Waltham, MA, USA) was used to aid the recovery of bacterial competent cells following transformation. In addition, 1M arabinose (Sigma, Saint Louis, MO, USA) was used to induce the expression of the Red genes. In order to speed up the induction step, it is necessary to assess the growth capacity on the arabinose of the strain. The host strain was streaked onto a M9 medium plate, supplemented with 1M arabinose and thiamine, and the plates were incubated overnight at 37 °C. If the strain does not ferment arabinose, induction of the helper plasmid can be performed simultaneously during the culture’s growth. In contrast, if the strain can use arabinose as a carbon source, the induction has to be carried out once the host strain has reached the appropriate growth level. 

MRC-5 cells are human lung fibroblasts (ATCC CCL-171), and HEK 293T cells are human kidney epithelial cells (ATCC CRL-11268), both of which were obtained from the American Type Culture Collection (ATCC; Manassas, VA, USA). All cells were maintained and grown in Dulbecco’s modified Eagle’s medium (DMEM, Gibco, Amarillo, TX, USA) containing 10% fetal bovine serum (FBS, Gibco, Amarillo, TX, USA), 20 mM glutamine (Lonza, Basilea, Suiza) and 10 units of penicillin and 10 µg of streptomycin (Lonza, Basilea, Suiza).

### 4.2. Plasmid DNA Purification

BACmid DNA for PCR amplification and transfection was obtained from *E. coli* using the NucleoBond BAC 100 Plasmid purification kit (Macherey & Nagel, Düren, Germany) according to the manufacturer’s instructions. Plasmid maintenance and amplification were performed from *E. coli* cells carrying pKD46 or pKD4 plasmid using the NZYMiniprep Kit (Nzytech, Lisboa, Portugal). The concentration of purified plasmid DNA preparations was determined by using a spectrophotometer BioPhotometer (Eppendorf, Hamburgo, Alemania).

### 4.3. Construction of the Parental Strain: Transformation of E. coli DH5α Containing the CMV BADrUL131-Y4 BAC with the Plasmid pKD46 

Transformation was performed by electroporation as previously described [58]. DH5α λ pir *E. coli* containing the CMV BADrUL131-Y4 BAC were streaked in a Chl-LB plate and incubated overnight at 37 °C. A single colony was inoculated in 5 mL Chl-LB and incubated overnight at 37 °C and 150 rpm. The next day, the overnight culture was diluted to a 0.1 OD_600_ in 25 mL of Cm-LB media and incubated at 37 °C with agitation (150 rpm) until reaching ~ 1.0 OD_600_ (3 h approximately). The culture was set on ice for 20 min to cool down, and all later steps were carried out at 4 °C, including the centrifugations. A total of 5 mL of bacterial culture were pelleted at 3900 g for 5 min to remove the LB media, and resuspended in 2 mL of cold HyPure Molecular Biology Grade water (Gybco, Amarillo, TX, USA). HyPure water washes were repeated three times in order to make electrocompetent cells. For that, bacterial pellets were resuspended in 2 mL, 1 mL and 500 µL Hypure cold water, respectively, followed by centrifugation at 13,000× *g* for 2 min. Supernatants were carefully removed with a pipette. After the last wash, the bacterial pellets were resuspended in 50 µL of Hypure cold water, and 50 ng of pKD46 plasmid were added. The mixture was incubated on ice for 5 min and passed to a pre-chilled 1mm electroporation cuvette (Biorad, Hercules, CA, USA). A negative control with no plasmid was also included. Electroporation was performed using a BioRad Gene pulser, and for the recovery of bacterial competent cells following transformation, 900 µL of SOC media were immediately added and were incubated for 2 h at 37 °C with no agitation. A total of 100 µL of the bacterial suspension were plated on Amp-Chl-LB plates and incubated at 30 °C overnight because pKD46 is a suicide plasmid at 37 °C. After incubation, transformant bacteria were stored at −80 °C using 900 µL of the bacterial culture with 20% glycerol. 

### 4.4. Preparation of the Linear DNA Kanamycin Resistance Cassettes 

We used a cassette containing the kanamycin resistance (Kan^R^) gene flanked by two sequences homologous to the deletion target gene [49]. Following the method developed by Datsenko and Wanner (2000), the Kan^R^ gene was amplified from the pKD4 plasmid, which contains the Kan^R^ gene flanked by two FRT sequences [48]. The forward and reverse oligonucleotides to amplify the Kan^R^ gene were designed to include specific sequences (17 and 18 bp, respectively) to amplify the Kan^R^ cassette, and at the 5′ end, each primer included a 50 bp tail homologous to the sequences flanking the genes to be deleted. Oligonucleotides were designed using the Ad169 HCMV strain (X17403.1). Both oligonucleotides had a Tm of 55 °C. PCR amplification of the Kan^R^ cassette was performed using EconoTaq PLUS (Lucigen, WI, USA). The PCR product was separated in a 1% agarose gel, and then the PCR product was cleaved from the gel and purified (GelPure, NzyTech, Lisboa, Portugal). These amplicons were sequenced using Sanger sequencing prior to the recombination process in order to check for random mutations.

### 4.5. Preparation of the Electrocompetent Cells 

To make electrocompetent cells, *E. coli* DH5α λ pir carrying the HCMV BADrUL131-Y4 BAC and the pKD46 helper plasmid were streaked in a Chl-Amp-LB plate and incubated overnight at 30 °C. The next day, a single colony was inoculated in 5 mL Amp-Chl-LB media and incubated overnight at 30 °C with agitation (150 rpm). It is important to use 30 °C because pKD46 is a suicide plasmid at 37 °C. The overnight culture was diluted to a OD_600_ of 0.1 in 100 mL of Amp-Chl-LB media and incubated at 30 °C and 150 rpm until reaching a 0.4 OD_600_ (approximately 4 h). To induce the expression of the pKD46 β, γ and exo genes, after the incubation, 1 mL of 1M Arabinose was added to the bacterial inoculum and incubated at 37 °C and 150 rpm until reaching a 1.2 OD_600_ (approximately one hour). The bacterial culture was next incubated on ice for 20 min, and all the following steps were performed at 4 °C. A total of 50 mL of pre-chilled cell culture was pelleted at 3900 g for 5 min to remove the LB media. Cells were then washed five times with HyPure water adding 20 mL, 10 mL, 2 mL, 1 mL and 500 µL of Hypure water, respectively. After each wash, cells were pelleted by centrifugation at 13,000× *g* for 2 min. Supernatants were carefully removed and discarded, and the pellet was carefully resuspended. After the last wash, the bacterial pellet was resuspended in 300 µL of Hypure cold water and distributed in three pre-chilled 1.5 mL tubes. Cells were either flash frozen in liquid nitrogen and stored at −80 °C for further use or freshly used for electroporation. 

### 4.6. Transformation of the E. Coli DH5α λ pir+BAC+pKD46 Strain with the Kan^R^ Cassette

Either freshly prepared or frozen electrocompetent stock cells kept on ice were used for transformation. A total of 250 ng of the target gene Kan^R^ cassette was added to cells, mixed and incubated on ice for 5 min. Electroporation was performed using the Gene Pulser Xcell Electroporation System (BioRad, Hercules, CA, USA). Right after electroporation, 500 µL of SOC were added and incubated 2 h at 37 °C without agitation for recovery. A total of 100 µL of the bacterial suspension were plated on a Kan-LB plate and incubated overnight at 37 °C. 

### 4.7. PCR Amplification and BAC Integrity

The colony PCR was performed to test the correct insertion of the Kan^R^ cassette in the target gene position. A single colony material, from the Kan-LB plate obtained after transforming the *E. coli* DH5α λ pir+BAC+pKD46 strain with the Kan^R^ cassette, was used in part to inoculate 5 mL of the Kan-LB medium that was grown overnight at 37 °C with agitation (150 rpm) for BAC plasmid purification. The rest of the single colony was added to 20 µL of NaOH 20 mM. The mixture was heated at 95 °C for 5 min and then centrifuged at 13,000× *g* for 1 min. In addition, 1 µL of the supernatant was used for PCR amplification. Three primer combinations were used, and primer sequences are shown in Table 3. One combination of oligonucleotides homologous to the upstream and downstream sequences of the deleted ORF was used (Table 1). For the other two PCR reactions, primer combinations with a homology to the Kan^R^ cassette combined with a primer either upstream or downstream of the deleted gene in the CMV genome, respectively, were employed. PCR products were separated by electrophoresis in a 1% agarose gel to check the product size. An overnight culture was used to purify the BAC DNA, as explained above. The PCR of the positive colonies was repeated using as a template of 0.5 ng of the purified plasmids.

To analyze the integrity of the BAC clones, 1500 ng of WT and ΔUL6-HCMV colonies were digested with *Eco*RI and *Bam*HI (FastDigest, ThermoFisher Scientific, Waltham, MA, USA). Digestion products were separated by electrophoresis in a 1% agarose gel to check the restriction pattern.

### 4.8. Cell Transfection

The day before transfection, 500,000 MRC-5 cells per well were seeded in a 6-well plate and were incubated for 24 h up to approximately 60% confluency. The HCMV BAC DNA was transfected in the cells using the Lipofectamine 2000 reagent (Invitrogen-ThermoFisher Scientific, Waltham, MA, USA) following the manufacturer protocol. Briefly, 4 μg of each BACmid DNA were diluted in 250 μL of OptiMEM and gently mixed. In parallel, 14 μL of Lipofectamine 2000 reagent were diluted in 250 μL of OptiMEM and incubated for 5 min at room temperature. Both the DNA mixture and Lipofectamine mixture were combined and incubated for 20 min at room temperature. A total of 500 μL of the DNA-lipofectamine complexes were added to each well and gently mixed by rocking the plate. Cells were then incubated at 37 °C in a CO_2_ incubator for 14 days. The cell medium was replaced by DMEM after 4 h. Virus production was monitored using a fluorescence microscope by the detection of GPF expression. 

### 4.9. Mutant Virus Propagation and PCR Verification

To demonstrate the generation of mutant viral progeny, the supernatants of wild type and mutant viral transfections were used to infect freshly seeded 90% confluent MRC-5 cells. After 14 days of the transfection, 15,000 MRC-5 cells were seeded in 96-well plates. Then, 24 h later, the medium was replaced with 100 µL of the different supernatants from the transfection wells (WT, ΔUL6-HCMV and ΔUL8-HCMV). After one hour of contact between the cells and the supernatants, the medium was again replaced by complete DMEM. Viral propagation was monitored by the detection of GPF expression by using a confocal SP5 Leica microscope.

Total DNA extraction was performed by using the E.Z.N.A. Tissue DNA Kit (Omega Biotek, Norcross, GA, USA). The oligonucleotides used are listed in Table 3, and the schematic representation of the process is shown in Figure 1A. 

## Figures and Tables

**Figure 1 ijms-22-10558-f001:**
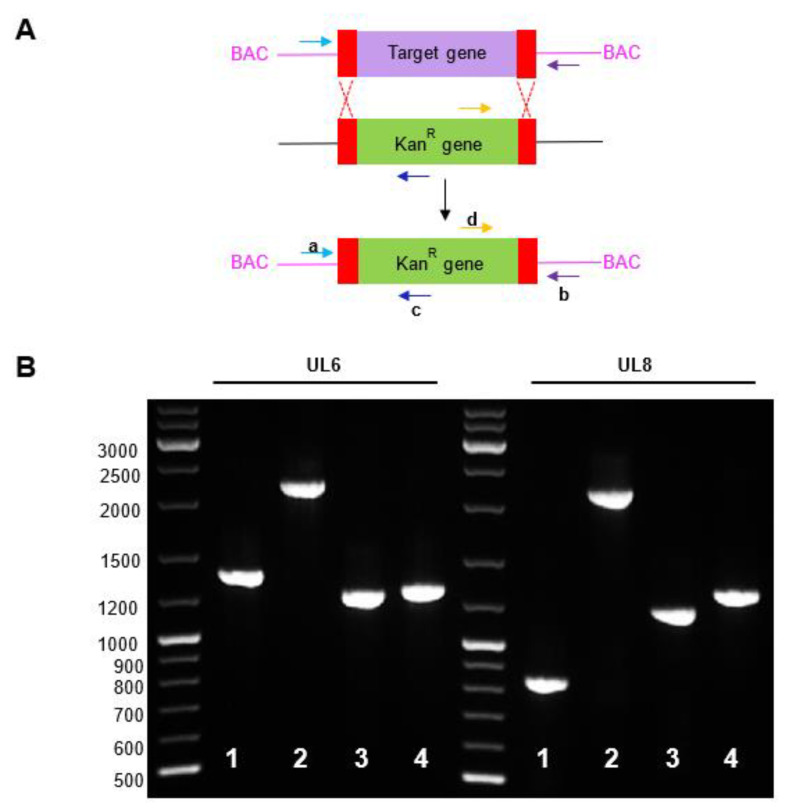
(**A**) Schematic representation of the homologous recombination system and the genomic location of the oligonucleotides used in the study. (**B**): PCR products obtained to confirm the indicated ΔUL6 and ΔUL8 gene deletion of the virus. PCR products were separated by electrophoresis in a 1.5% Agarose gel. Lane 1: amplification using the primers upstream and dowstream (ab) of the target gene using the wild-type BAC DNA (WT) as a template. Lane 2: amplification using the primers upstream and dowstream (ab) of the target gene using the indicated deletion mutant BAC DNA as a template. Lane 3: amplification using the primer upstream of the deleted gene and an internal Kan primer (ac). Lane 4: amplification using the internal Kan primer and a primer downstream of the deleted gene (db). One kb DNA molecular weight marker was used as a ladder.

**Figure 2 ijms-22-10558-f002:**
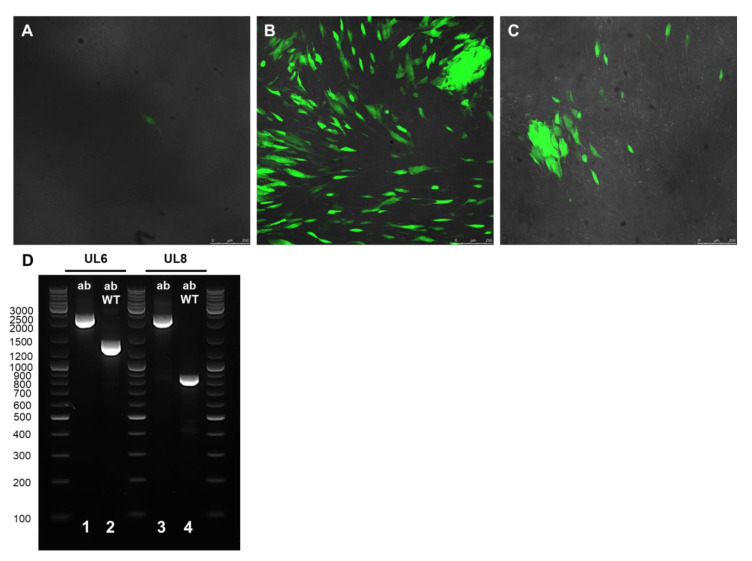
Representative images of MRC-5 cells infected with WT HCMV-BAC (**A**), ΔUL6-HCMV-BAC (**B**) and ΔUL8-HCMV-BAC (**C**) 14 days post-infection. Infected cells express GFP protein due to the integration of the gene in the BAC genome. (**D**) PCR products obtained to confirm the indicated ΔUL6 and ΔUL8 gene deletion of the reconstituted virus after the transfection in MRC-5 cells. PCR products were separated by electrophoresis in a 1.5% Agarose gel. Lane 1: amplification using the primers upstream and dowstream (ab) of the target gene using the indicated deletion mutant BAC DNA as a template. Lane 2: amplification using the primers upstream and dowstream (ab) of the target gene using the wild-type BAC DNA (WT) as a template. One kb DNA molecular weight marker was used as a ladder.

**Figure 3 ijms-22-10558-f003:**
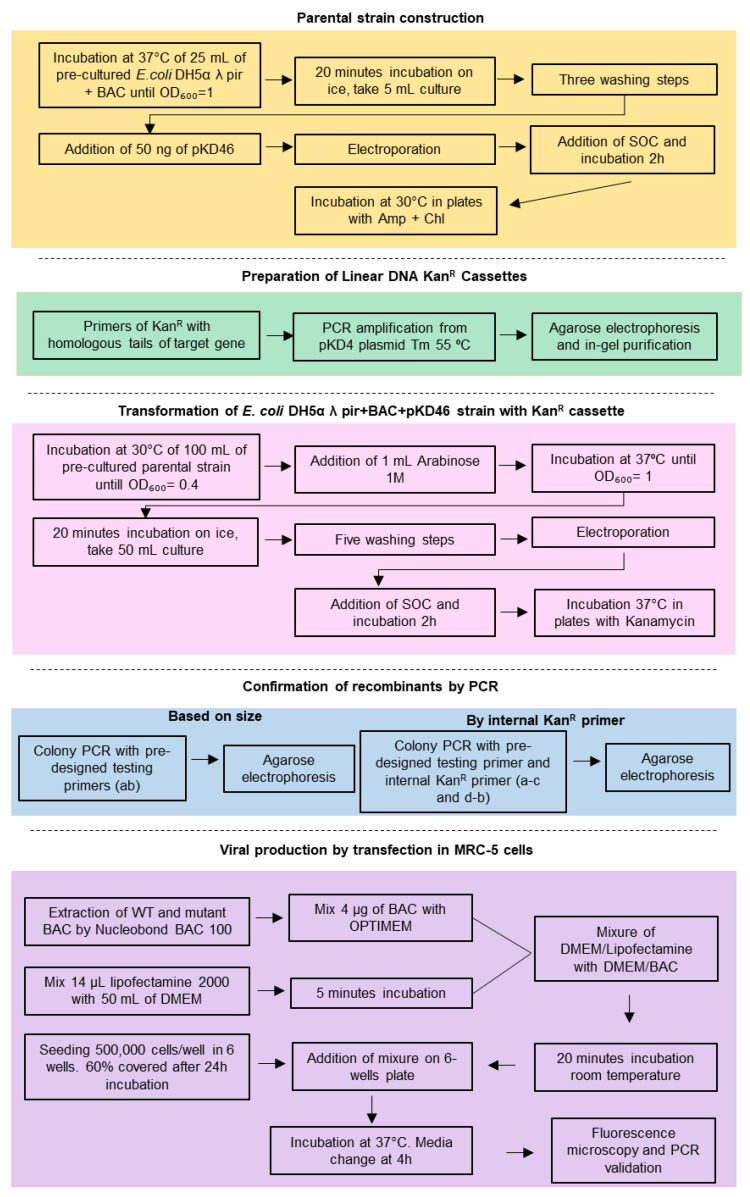
Workflow representation of the methodology for the recombination system including the experimental procedures.

**Table 1 ijms-22-10558-t001:** Colonies obtained for the indicated deleted genes. The number of positive colonies for the first and second set of colonies tested and the percentage of positive colonies are represented for each of the deleted gene.

Gene	Number of Tested Colonies	Positive Colonies (Test 1)	Positive Colonies (Test 2)	Percentage of Positive (%)
UL2	20	4	4	20
UL5	4	1	1	25
UL6	7	1	1	14.3
UL7	4	2	2	50
UL8	16	2	2	12.5
UL9	24	1	1	4.2
UL44	19	2	2	10.5
UL121	55	1	1	1.8
US7	10	6	6	60
US9	42	21	21	50
US13	10	1	1	10
US14	4	3	3	75
US16	10	5	5	50
US17	4	3	3	75
US18	22	6	6	27.2
US20	15	11	11	73.3
US21	15	7	7	46.7
US30	15	1	1	6.7
pp65	15	3	3	20
IE1	15	1	1	6.7
gO	15	7	7	46.7
gM	6	6	6	100

**Table 2 ijms-22-10558-t002:** Oligonucleotides used for amplification of Kanamycin resistance gene and mutant validation.

Oligonucleotides	Sequence (5′–3′)
UL6KanR-F (a)	ATGCATGCTAAGATGAACGGGTGGGCTGGGGTGCGCTTGGTAACTCACTGGCATTACACGTCTTGAG
UL6KanR-R (a)	CTACATTAACAAACCACGTTCTTCATCGTCCACGTGGCTTCGCCAGCGTCTAAGGTTTAACGGTTGTG
UL8KanR-F (a)	ATGACTAACCCTGGGCTATATGCATCGGAAAATTATAACGGAAATTATGAGCATTACACGTCTTGAG
UL8KanR-R (a)	TCACAGCTCCGTGTCCGTCATAAATACTTGTCCGTACTGTTTATTGTCTTTAAGGTTTAACGGTTGTG
UL6KanRcp-F (b)	TCCATCTGTCGTTTCTG
UL6KanRcp-R (b)	CCTCTCCACGTTTGTAA
UL8KanRcp-F (b)	GTTTAGCACCACAACTAC
UL8KanRcp-R (b)	CTGGCTTGCTATCTATTT
KanRInt-F (c)	GGATCTCCTGTCATCTC
KanRInt-R (c)	CATGATATTCGGCAAGC
Seq-F (d)	GCATTACACGTCTTGAG
Seq-R (d)	TAAGGTTTAACGGTTGTG

(a) Oligonucleotide used for amplification of the knockout cassette. The sequence with homology to the kan^R^ cassette from the plasmid pKD4 is underlined. The remainder of the primer sequence is homologous to the flanking region of the deleted ORF. (b) Oligonucleotide used to check the correct insertion of the deletion cassette (c) Oligonucleotide used for a second check whose sequence is a part of the Kan^R^ gene (d) Oligonucleotide used for Sanger sequencing.

**Table 3 ijms-22-10558-t003:** Comparison of the results using freshly prepared (Fresh) versus frozen electrocompetent cells for each of the indicated genes. The electroporation was performed at 1.8 volts, and the time constant (Time) and the number of colonies obtained for each transformation are indicated.

	UL8		US20		US21		gM		gO		IE1		pp65	
	Fresh	Frozen	Fresh	Frozen	Fresh	Frozen	Fresh	Frozen	Fresh	Frozen	Fresh	Frozen	Fresh	Frozen
Time (ms)	4.9	4.7	4.5	3.2	4	3.3	4.3	3.8	4.3	3.7	4.2	3.9	4.2	3.6
Number of total colonies	16	190	43	52	27	73	10	21	34	31	26	52	47	23
Number of tested colonies	16	16	15	15	20	20	6	6	17	17	15	15	16	16
Number of positive colonies	2	11	11	12	9	17	6	6	8	13	1	5	3	9
Percentage of positive	12.5	68.75	73.33	80	45	85	100	100	47.05	76.47	6.66	33.33	18.75	56.25

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
