# Peer review of "Optimization of a Lambda-RED Recombination Method for Rapid Gene Deletion in Human Cytomegalovirus"

_ijms, 2021, doi:10.3390/ijms221910558_

Round 1

Reviewer 1 Report

discussion:
on lines 187 to 191, you discuss the use of sequencing of PCR products, but on this paper you didn't used this technique to confirm any possible mutations, explained why you didn't in the discussion.

material and methods:
change rpm to the corresponding g-force on lines: 292, 294, 297, 338, 341, 348, 374.

Author Response

Thank you for your comments and for those of the reviewers regarding our recently submitted manuscript. We appreciate the time and effort that have gone into reviewing our manuscript and for the constructive comments of the reviewers. We have modified the paper, taking into account both the reviewers’ comments. A point-by-point response is included below.

We hope that the manuscript is now acceptable for publication.

We look forward to your response.

Reviewer 1

  1. discussion:
    on lines 187 to 191, you discuss the use of sequencing of PCR products, but on this paper you didn't used this technique to confirm any possible mutations, explained why you didn't in the discussion.

Response: We appreciate the reviewer’s comment. In fact, we sequenced the PCR products and in order to clarify this, we have included a paragraph in the material and methods section (lines 348-350) in order to address this point.

  1. Material and methods:
    change rpm to the corresponding g-force on lines: 292, 294, 297, 338, 341, 348, 374.

Response: Following the reviewer’s suggestion, we have included g-force in lines 297 and 348 where samples were spun down using centrifuges. However, the information included in Lines 294, 338 and 375 refers to the use of an orbital incubator using 150 rpm for agitation. We have included this information to clarify this fact.

Reviewer 2 Report

In their current manuscript, Garcia-Rios et al. present data on the modification of HCMV BAC DNA by lambda-RED recombineering, which they used to make multiple single gene deletions. The effect of the gene deletions are not studied in this manuscript, however, and the authors show only the technical part of the recombineering process.

What exactly constitutes the "optimized" protocol? The overall approach seems rather standard lambda recombineering to me? The authors should elaborate more on how their approach differs from standard technique. What are the crucial steps of the optimization and how much do they improve the process?

The authors give the frequency of positive and negative clones, but it would also be interesting to get some more information about the negative clones. The clones were selected on Kanamycin, so did the bacteria not lose the pKD46 plasmid, or was there insertion of KanR in a wrong region? How about the integrity of the whole BAC? A gel with digested BAC DNA for a larger number of positive and negative clones should be shown. Can all clones that were positive by PCR be used for reconstitution of virus?

Similarly, I wonder how the efficiency of deletion compares to the efficiency of traditional deletion by homologous recombination in the authors' hands? A side by side comparison for one gene deletion would be interesting and informative.

There is a legend to Figure 3 but no actual figure 3 is contained in the manscript.

Author Response

Response to reviewer:

Thank you for your comments and for those of the reviewers regarding our recently submitted manuscript. We appreciate the time and effort that have gone into reviewing our manuscript and for the constructive comments of the reviewers. We have modified the paper, taking into account both the reviewers’ comments. A point-by-point response is included below.

We hope that the manuscript is now acceptable for publication.

We look forward to your response.

Reviewer 2

In their current manuscript, Garcia-Rios et al. present data on the modification of HCMV BAC DNA by lambda-RED recombineering, which they used to make multiple single gene deletions. The effect of the gene deletions are not studied in this manuscript, however, and the authors show only the technical part of the recombineering process.

  1. What exactly constitutes the "optimized" protocol? The overall approach seems rather standard lambda recombineering to me? The authors should elaborate more on how their approach differs from standard technique. What are the crucial steps of the optimization and how much do they improve the process?

Response: Following reviewer’s suggestion we have introduce a paragraph to specify the critical steps and the novelty of the optimized method (Lines 215-221). Specifically, following the protocol published by Datsenko and Wanner we observed that 85% of clones that were obtained were negative (did not contain the expected mutation). In the original paper, to make electrocompetent cells they used an OD of 0.6 however, an OD of 1.0 was the optimum for manipulating the CMV bacmid. In addition, they used between 10-100ng of the PCR product for E. coli transformation, while we have increased this parameter up to 250 ng of PCR product allowing us to obtain all the tested deletions. Furthermore, in the original work arabinose induction was introduced in the culture when initiating growth however the number of positive colonies was increased when cells were induced when the culture reach an OD600 of 0.4.

  1. The authors give the frequency of positive and negative clones, but it would also be interesting to get some more information about the negative clones. The clones were selected on Kanamycin, so did the bacteria not lose the pKD46 plasmid, or was there insertion of KanR in a wrong region?

Response: In response to the reviewer’s question, the plasmid pKD46 was lost since we grew all the tested colonies in ampicillin plates and they were unable to grow, whereas they grew perfectly using kanamycin for selection. We believe that in the negative clones, the KanR cassette was integrated in the wrong region in the genome however, negative colonies were discarded and were not further analyzed.

  1. How about the integrity of the whole BAC? A gel with digested BAC DNA for a larger number of positive and negative clones should be shown. Can all clones that were positive by PCR be used for reconstitution of virus?

Response: In response to the reviewer’s suggestion, we have performed the digestion of WT, DUL6 and DUL8 BACs using EcoRI and BamHI to test DNA integrity of the entire BAC. The results are shown in Supplementary Figure 1, and a paragraph including this experiment was added in the result section (Lines 153-160) and in the Material and Methods section (Lines 421-424). As explained in the previous point, we did not further analyzed the negative colonies, thus we cannot include results from the negative clones.

In addition, for a subset of deletion mutants we have rescued viable virus progeny after transfection, indicating that the viral genome is intact.

Similarly, I wonder how the efficiency of deletion compares to the efficiency of traditional deletion by homologous recombination in the authors' hands? A side by side comparison for one gene deletion would be interesting and informative.

Response: We agree with reviewer comment. However, we performed the optimization of the method after several unsuccessful attempts. However, in our hands the efficiency of the original method was very low obtaining in all cases between none to three transformants, approximately15% efficiency. After optimization of the method, we had 100% of efficiency even in those cases in which the number of colonies obtained after the transformation was low. We have include this information in the discussion (Lanes 215-217).

  1. There is a legend to Figure 3 but no actual figure 3 is contained in the manuscript.

Response: Figure 3 has been included in the manuscript.

Reviewer 3 Report

In this paper, the authors report the optimization of a method to obtain gene deletions in the HCMV genome. In particular, they describe the Lambda-Red Recombination system, based on the BAC technology.

Overall, the experimental approach was seemed to run correctly, and the analyses look right. The topic is relevant and the development of new methods to manipulate the HCMV genome is absolutely needed. The manuscript is well-written and organized. The figures are adequate.

However, since the Lambda-Red Recombination system is a method already available and different papers employed this system, it is crucial to underline in the discussion the differences in the method used by the authors compared to the previously published papers that allowed the improvements claimed by the authors. Otherwise, it is not clear which is the novelty of the paper.

Another important point is to prove if the method is suitable also for the insertion of point mutations instead of short or long deletions.

A minor point: many general references about HCMV biology and genome are quite dated. They should be replaced or better integrated with more updated references.

Author Response

Response to reviewers:

Thank you for your comments and for those of the reviewers regarding our recently submitted manuscript. We appreciate the time and effort that have gone into reviewing our manuscript and for the constructive comments of the reviewers. We have modified the paper, taking into account both the reviewers’ comments. A point-by-point response is included below.

We hope that the manuscript is now acceptable for publication.

We look forward to your response.

Reviewer 3

In this paper, the authors report the optimization of a method to obtain gene deletions in the HCMV genome. In particular, they describe the Lambda-Red Recombination system, based on the BAC technology.

Overall, the experimental approach was seemed to run correctly, and the analyses look right. The topic is relevant and the development of new methods to manipulate the HCMV genome is absolutely needed. The manuscript is well-written and organized. The figures are adequate.

  1. However, since the Lambda-Red Recombination system is a method already available and different papers employed this system, it is crucial to underline in the discussion the differences in the method used by the authors compared to the previously published papers that allowed the improvements claimed by the authors. Otherwise, it is not clear which is the novelty of the paper.

Response: Following reviewer’s suggestion, we have introduced a paragraph to specify the critical steps and the novelty of the optimized method (Lines 215-221). Specifically, following the protocol published by Datsenko and Wanner we observed that 85% of clones that were obtained were negative (did not contain the expected mutation). In the original paper, to make electrocompetent cells they used an OD of 0.6 however, an OD of 1.0 was the optimum. In addition, they used between 10-100ng of the PCR product for E. coli transformation, while we have increased this parameter up to 250 ng of PCR product allowing us to obtain all the tested deletions. Furthermore, in the original work arabinose induction was introduced in the culture when initiating growth, however the number of positive colonies was increased when cells were induced when the culture reach an OD600 of 0.4.

  1. Another important point is to prove if the method is suitable also for the insertion of point mutations instead of short or long deletions.

Response: In response to the reviewer’s comment, as shown in the results section we were able to replace the deleted gene by a cassette containing the KanR gene. This method could be applicable for the insertion of point mutations including these differences during the design of the cassette. The successful insertion of the point mutation could be easily detected by sequencing. To clarify this issue, we have add a sentence in the discussion section (Lines 225-227).

  1. A minor point: many general references about HCMV biology and genome are quite dated. They should be replaced or better integrated with more updated references.

Response: Following the reviewer´s comment we have introduced new updated references in the introduction. 

Round 2

Reviewer 2 Report

Following my previous suggestion, the authors added a figure showing an agarose gel of EcoRI, BamHI restricted BACs. I think my initial query would be better answered by showing a gel of tested clones from one recombineering, not just one positive clone. The same is actually true for the agarose gel showing the PCR results. Surely the authors did not analyse only one clone but screened more, since they also say that not all clones were positive (line 111: 1.8 to 100 %). More data should be shown, not just data of one selected positive clone for two exemplary constructs.

With regard to the gel shown, I personally would not be able to judge the integrity of the BACs from a gel that shows most bands as one big bright running together band. The number and size of bands clearly calls for a low-percent gel and long separation, not 1.5% as shown here.

Regarding the overall efficiency of the recombineering approach, in the discussion the authors suddenly claim an efficiency of 100 %, directly contradicting the previous statement, compare lines 225 vs. 111. Also in the abstract, the authors also talk about 100 % efficiency (line 19-20: "100% efficiency after transfection into eukaryotic cells"), but apparently referring to the reconstitution of the recombineered BAC. Since getting the recombineered / recombinant BAC in most cases is the harder part, the wording in the abstract is misleading and should be changed. 

Finally, some of the modifications the authors made to the protocol have also been used by others before, so the overall novelty of the approach is not as great, although it may be in the CMV field. Other publications with similar approaches in other virus systems should be referenced.

There seems to be a mistake in figure 3 where in the third box, there are two boxes indicating incubation on kanamycin plates, surely the bacteria were not put on plates between washing and electroporation?

At the moment, the authors still show only a minimal amount of original data and make overstated and contradictory claims on the efficiency of the presented method. If further analyses were performed with the obtained BACs, I would probably agree that not all steps of the production should be shown in all detail. But since this manuscript is only on the technical side of producing these mutant BACs, all steps should be shown in great detail and with all technical information and as much accuracy as possible.

I am therefore not able to endorse this manuscript for publication.

Author Response

We appreciate the constructive comments of the reviewer. We have modified the paper, taking into account the reviewer’s comments. A point-by-point response is included below.

We hope that the manuscript is now acceptable for publication.

We look forward to your response.

Reviewer 2

Following my previous suggestion, the authors added a figure showing an agarose gel of EcoRI, BamHI restricted BACs. I think my initial query would be better answered by showing a gel of tested clones from one recombineering, not just one positive clone. The same is actually true for the agarose gel showing the PCR results. Surely the authors did not analyse only one clone but screened more, since they also say that not all clones were positive (line 111: 1.8 to 100 %). More data should be shown, not just data of one selected positive clone for two exemplary constructs.

With regard to the gel shown, I personally would not be able to judge the integrity of the BACs from a gel that shows most bands as one big bright running together band. The number and size of bands clearly calls for a low-percent gel and long separation, not 1.5% as shown here.

In response to the reviewer’s suggestion, we have performed a new digestions with EcoRI and BamHI enzymes using DNA from the WT BAC and the BAC obtained from 8 colonies. Digestion products were separated by electrophoresis on a 1% agarose gel.  The restriction patterns obtained are shown in Supplementary Figure 1.  In addition, we includedtwo panels showing the two rounds of PCR performed to validate the insertion of the Kan resistance cassette in the positive clones in Supplementary Figure 1. A paragraph including this experiment was added in the result section (Lines 153-166).

Regarding the overall efficiency of the recombineering approach, in the discussion the authors suddenly claim an efficiency of 100 %, directly contradicting the previous statement, compare lines 225 vs. 111. Also in the abstract, the authors also talk about 100 % efficiency (line 19-20: "100% efficiency after transfection into eukaryotic cells"), but apparently referring to the reconstitution of the recombineered BAC. Since getting the recombineered / recombinant BAC in most cases is the harder part, the wording in the abstract is misleading and should be changed. 

Following the reviewer suggestion we have clarified that improving the method has allowed us to obtain positive colonies for all the tested genes. We have modified the manuscript accordingly (Abstract and Lines 19-20).

Finally, some of the modifications the authors made to the protocol have also been used by others before, so the overall novelty of the approach is not as great, although it may be in the CMV field. Other publications with similar approaches in other virus systems should be referenced.

In response to the reviewer´s comment we have optimized the method to improve the efficiency of the protocol that has been previously used mostly for bacteria and only few virus systems. Following reviewer’s suggestion, a paragraph including references to other viral systems has been included (Lines 220-222).

There seems to be a mistake in figure 3 where in the third box, there are two boxes indicating incubation on kanamycin plates, surely the bacteria were not put on plates between washing and electroporation?

The reviewer is right. We have corrected figure 3.

At the moment, the authors still show only a minimal amount of original data and make overstated and contradictory claims on the efficiency of the presented method. If further analyses were performed with the obtained BACs, I would probably agree that not all steps of the production should be shown in all detail. But since this manuscript is only on the technical side of producing these mutant BACs, all steps should be shown in great detail and with all technical information and as much accuracy as possible.

In response to the reviewer’s comment we believe that the topic is relevant and the development of new methods to manipulate the HCMV genome is needed. Generating gene deletions in large viral genomes (such as HCMV) continues to be a challenge that limits experimental characterization of gene function in these viruses.  We agree with the reviewer that the optimized method that we are reporting builds upon previous work.  However, we feel that the innovations introduced in our study will provide a valuable resource for researchers aiming to efficiently generate gene deletions in large viral genomes. Following the reviewer’s comment we have introduce a paragraph to highlight the importance of this work (Lines 230-233, 275-277).

Reviewer 3 Report

The answers provided by the authors are appropriate. In my opinion, no additional editing is required and the paper can be accepted.

Author Response

We appreciate the constructive comments of the reviewer.